# Preparation, Characterization, and Anti-Adhesive Activity of Sulfate Polysaccharide from *Caulerpa lentillifera* against *Helicobacter pylori*

**DOI:** 10.3390/polym14224993

**Published:** 2022-11-18

**Authors:** Bao Le, Duy Thanh Do, Hien Minh Nguyen, Bich Hang Do, Huong Thuy Le

**Affiliations:** 1Faculty of Pharmacy, Ton Duc Thang University, Ho Chi Minh City 700000, Vietnam; 2Faculty of Applied Sciences, Ton Duc Thang University, Ho Chi Minh City 700000, Vietnam

**Keywords:** *Caulerpa lentillifera*, gastric inflammation, *Helicobacter pylori*, polysaccharide

## Abstract

In the gastric mucosa, chronic inflammation due to *Helicobacter pylori* infection promotes gastrocarcinogenesis. Polysaccharides of *Caulerpa lentillifera* are well-characterized by broad antimicrobial activity and anti-inflammatory potentials. The present study was undertaken to investigate whether the low molecular sulfate polysaccharides of *C. lentillifera* (CLCP) exhibit any anti-adhesive activity against *H. pylori*. After a hot water extraction and purification process, two purified polysaccharide fractions (CLCP-1 and CLCP2) were studied based on structural characterization and bioactivity determination. The results implied that except for the molar ratio, CLCP-1 and CLCP-2 contain high sulfate, mannose, galactose, xylose, glucose levels, and low protein levels. The molecular weight and Fourier transform infrared spectroscopy (FT-IR) assays confirmed that CLCP-1 and CLCP-2 are sulfate polysaccharides with an average molecular weight (Mw) of 963.15 and 648.42 kDa, respectively. In addition, CLCP-1 and CLCP-2 exhibited stronger antibacterial activity against H. pylori. CLCP-1 and CLCP-2 could significantly promote macrophage proliferation and decrease the production of nitric oxide (NO) through downregulated expression of inducible nitric oxide synthase (iNOS). Meanwhile, CLCP-1 and CLCP-2 in this study showed efficiently protected gastric adenocarcinoma (AGS) cells against *H. pylori* with the inhibition of the IL-8/NF-κB axis. These findings suggested the effect of *Caulerpa lentillifera* polysaccharides on *H. pylori* adhesion, a potential supply of nutrients for eradication therapy through the reduction of cell count and inflammation.

## 1. Introduction

*Helicobacter pylori* is a true pathogen in the pathogenesis of gastritis and peptic ulcer disease. Left untreated, it can cause prolonged gastric inflammation leading to stomach cancer [1]. Globally, the prevalence of *H. pylori* infection accounts for more than 50%, with a much higher prevalence in developed countries ranging from 70–90% [2]. Currently, triple or quadruple therapies, including a proton pump inhibitor (PPI), two antibiotics, and treatment with or without bismuth, are the most used treatments [3]. Despite elimination rates of 60% to 90%, some concerns persist as the rise of antibiotic resistance (with high divergence) and the effectiveness of current regimens have diminished over the years. Since 2017, the World Health Organization (WHO) has announced that the prevalence of resistance of *H. pylori* to clarithromycin and metronidazole is over 15% [4]. Clarithromycin is a second-generation macrolide and the most potent antibiotic in *H. pylori* eradication treatment regimens [5]. Clarithromycin inhibits the protein synthesis of *H. pylori* by targeting the 50S ribosomal subunit [6]. Metronidazole is a synthetic nitroimidazole derivative activated by nitro-reductase to produce oxygen radicals toxic to bacteria through DNA damage [7]. Metronidazole-resistant *H. pylori* may be caused by mutations in rdxA, which encodes oxygen-insensitive NADPH nitroreductase [8]. Following the rapid development of antibiotic resistance, the other drawbacks of these treatment failures are antibiotic degradation by the acidic stomach environment, use of non-FDA-approved agents (e.g., tetracycline and nitazoxanide), severe adverse effects, and poor patient compliance [9,10]. Taken together, it is necessary to develop alternative therapeutic approaches for the prevention or/and treatment of *H. pylori* infections. Diverse alternative treatment options have been underinvestigated, including those potentially used in clinical practice, such as probiotics and chemotherapy, photodynamic therapy, natural sources, vaccines, nanoparticles, and probiotic/prebiotic therapy [10,11,12].

*Caulerpa lentillifera* (Bryopsidophyceae, Chlorophyta) is an edible green seaweed with various beneficial effects on human health [13]. Indeed, *C. lentillifera* contains numerous biologically active natural products, including siphonaxanthin, phenolic, and polysaccharide [14]. Sulfate polysaccharides isolated from *C. lentillifera* (CLCP) have been thoroughly described in earlier studies and exert antiviral [15], antioxidant [16], immunostimulatory [17], anti-inflammatory [18], anti-diabetic [19], and anticancer activities [20]. Recent studies have indicated that polysaccharides obtained from various sources have effectively enhanced protective immunity against *H. pylori* by inhibiting the adhesion of *H. pylori* and reducing the inflammatory response of a gastric epithelial cell to *H. pylori* [21]. A sulfate polysaccharide with a molecular mass >8 kDa has been shown to enhance the immunostimulatory effects [22]. In addition, the polysaccharide containing a high sulfate content (21.26%) could possess anti-inflammatory solid activities [18]. Recurrent discoveries suggest CLCP would be a promising alternative to anti-*H. pylori*, although its mechanism remains underexplored thus, there is a need to further exploitation of CLCP.

Several studies have shown that a polysaccharide’s molecular weight is a crucial factor in its antimicrobial properties [23]. Therefore, in the present study, we prepared CLCP fractions with a low molecular weight (<1000 kDa) and evaluated the antimicrobial potential of CLCP fractions against *H. pylori;* further investigations were conducted to ascertain the mechanisms underlying the protective effects of CLCP fractions against *H. pylori* in mouse macrophage and gastric adenocarcinoma cells. These results could contribute to further exploitation of the production of supplements for *Helicobacter pylori* eradication by *Caulerpa lentillifera* polysaccharides.

## 2. Materials and Methods

### 2.1. Bacterial Strains and Culture Condition

The reference *H. pylori* KTCC 12083 and the other three clinical isolated strains HP 43504, HP 51932, HP 700392 used in the present study were kindly provided by the Oxford University Clinical Research Unit (OUCRU) in Ho Chi Minh City (HCMC). The lyophilized culture was revived according to the protocol provided by the KTCC. Cells were grown on Brucella Agar plates (BD BBL™ Brucella Agar, Difco™, Sparks, MD, USA) supplemented with 7% (*v*/*v*) sheep blood (BA-SB) for 5–7 days under microaerobic conditions (O_2_ 5%; CO_2_ 15%; N_2_ 85%) at 37 °C using a GasPak™ EZ Anaerobe Container system (Difco™, Sparks, MD, USA). After incubation, cells harvested by centrifugation were washed three times with 0.1 M PBS (pH 7.4) before resuspension. The final concentration of the culture stock was 1 ± 0.2.6 × 10^8^ CFU/mL.

### 2.2. Polysaccharide Extraction

*Caulerpa lentillifera* was purchased from a local aquatic farm (Nha Trang, Khanh Hoa Province, Vietnam). The polysaccharide was prepared with slight alterations based on our previous method [24]. Briefly, *C. lentillifera* powders were extracted with hot water for 3 h at 100 °C, and a precipitate was obtained by adjusting the ethanol concentration in the solution stepwise to 75%. The precipitate was washed three times with dry ethanol, lyophilized, and then deproteinized with the Sevag method. The concentrated extract was dialyzed using 2K MWCO Slide-A-Lyzer™ dialysis flasks (Thermo Fisher Scientific, Inc., Waltham, MA, USA) against distilled water for 40 h. The outer supernatant of the ultra-filtration membrane was collected, concentrated by rotary evaporation at 50 °C, and lyophilized to prepare crude polysaccharide (CLCP). The CLCP was fractionated by anion-exchange chromatography using DEAE-Sephadex A-25 (GE Healthcare, Germany) and using gradually increasing concentrations of sodium chloride (0–2.5 M) at a flow rate of 1 mL/min, and eluents (5 mL/tube), giving two subfractions CLCP1 and CLCP2. Then the polysaccharide fractions were concentrated and dried by freeze-drying.

### 2.3. Chemical Composition Analysis

The total sugar, sulfate, uronic acid, and protein contents of the CLCP fractions were measured by the phenol-sulfuric acid method using mannose as a standard [25], the barium chloride-gelatin method using K_2_SO_4_ as standard [26], the carbazole-sulfuric acid method using D-glucuronic acid as standard [27], and a Pierce™ BCA protein assay kit (Thermo Fisher Scientific, Rockford, IL, USA), respectively.

The molecular weights (Mw) of the CLCP fractions were analyzed by a high-performance gel permeation chromatography (HPGPC, Agilent 1100) instrument equipped with a refractive index detector (RID-10A, Shimadzu, Tokyo, Japan). The polysaccharide solution (15 mg/mL) was injected through a TSK-GEL G4000SW_XL_ column (300 mm × 7.8 mm, Tosoh Co., Tokyo, Japan) and eluted with deionized water at a flow rate of 0.6 mL/min. The calibration curve was plotted using the retention times of standard dextrans (10–2000 kDa, Sigma-Aldrich, MO, USA).

Monosaccharide composition analysis was carried out based on pre-column derivatization with 1-phenyl-3-methyl-5-pyrazolone (PMP) and analyzed by HPLC essentially as described in our previous paper [28].

The zeta (ζ)-potential was determined by Zetasizer NANO ZS-ZEN3600 (Malvern Instruments Ltd., Worcestershire, UK) at 25 °C through the laser beam operated at 659 nm to estimate the surface charge of CLCP fractions.

FTIR analysis was carried out by a VERTEX 70 v Fourier transform infrared spectrometer (Bruker, Ettlingen, Germany) to characterize the functional groups of CLCP fractions. The analytical powders were ground with KBr and then pressed into KBr pellets, which were then detected in the wave number range of 400–4500 cm^−1^.

### 2.4. Antimicrobial Activity Assays

For the evaluation of antimicrobial activity by broth micro-dilution methods, 50 μL of each *H. pylori* bacterial solution (1 ± 0.2.6 × 10^8^ CFU/mL) was added to 96 microtiter plate wells containing 100 μL of Brucella broth (BD BBL™ Brucella Agar, Difco™, Sparks, MD, USA) supplemented with 7% (*v*/*v*) sheep blood and CLCP fractions at final concentrations of 62.5, 125, 250, 500, and 1000 µg/mL. Brucella broth containing 7% (*v*/*v*) sheep blood (BB-SB) was used as the control broth. The plates were incubated in microaerophilic conditions with moderate agitation at 37 °C. Aliquots were taken over time and the OD600 readings were measured at 48 h. After measuring absorbance, 10 µL of each well was spread-plated on BA-SB agar plate and incubated at 37 °C for 5 days to verify microbial growth.

### 2.5. Cell Line, Cell Culture, and Cell Viability Assay

Mouse macrophage cells RAW 264.7 and human gastric adenocarcinoma AGS cells were provided by Korean Cell Line Bank (Seoul, Korea). RAW 264.7 cells were cultured in DMEM (WelGene, Gyeongsan, Gyeongsangbuk-do, Korea) with 10% fetal bovine serum (FBS; WelGene) at 37 °C with 5% CO_2_. AGS cells were grown in RPMI 1640 medium containing 25 mM HEPES supplemented with 10%, 50 units/mL penicillin, and 50 μg/mL streptomycin (WelGene) at 37 °C with 5% CO_2_.

RAW 264.7 and AGS cells were seeded into 24-well tissue culture plates (Corning CellBIND Surface, Corning, NY, USA) at a density of 5 × 10^4^ cells/mL and incubated at 37 °C in a 5% CO_2_ atmosphere. The cells were treated with 31.25, 62.5, 125, 250, 500, and 1000 μg/mL of CLCP fractions. After incubating for 24 h at 37 °C with 5% CO_2_, RAW 264.7 cells were then stimulated with LPS (1.0 μg/mL) for 6 h, supernatants were collected and the cells were washed three times with PBS. Then, 10 μL of MTT solution (5 mg/mL) was added to each well. After incubating for 4 h, the supernatant from the wells was carefully discarded, and 100 μL dimethyl sulfoxide was added. The optical density was evaluated at 490 nm by the multi-plate reader in three replicates.

### 2.6. Measurement of NO Production and iNOS Expression from RAW 264.7 Cells

RAW 264.7 cells were plated at 4 × 10^5^ cells/well in 24-well plates and then incubated with 31.25, 62.5, 125, 250, 500, and 1000 μg/mL of CLCP fractions in the presence of LPS (10 μg/mL) for 24 h in a complete DMEM medium. Then, the supernatant medium of 100 μL in each group was transferred and mixed with 100 μL of Griess reagent (Sigma) in 96-well plates and incubated at room temperature for 10 min. Absorbance was measured at 546 nm using a spectrophotometer. Fresh DMEM medium were used as blanks in all experiments. NO levels in the samples were read off a standard sodium nitrite curve.

RAW 264.7 cells were seeded at 4 × 10^5^ cells/well in 24-well plates and then incubated with 31.25, 62.5, 125, 250, 500, and 1000 μg/mL of CLCP for 2 h, and then stimulated with LPS (100 μg/mL) for 24 h. The mouse inducible nitric oxide synthase (iNOS) activity in the treated cells was detected with an iNOS sandwich ELISA kit (ab253219, Abcam, Cambridge, MA, USA) and the absorbance was then measured at 450 nm using a multi-plate reader.

### 2.7. Helicobacter pylori Adherence Assay

AGS cells were seeded (~5 × 10^4^ cells/well) in RPMI 1640 medium containing no antibiotics in 24-well plates at 37 °C with 5% CO_2_ until a monolayer was formed. When the cell cultures were 70–80% confluent, the pre-, co-, and post-incubation experiments were performed as follow: (a) in the pre-incubation experiment, a monolayer was incubated with CLCP fractions for 2 h, then washed three times with PBS. The HP 700392 *H. pylori* suspension was added to infect the AGS cell for 2 h; (b) in the co-incubation experiment, a monolayer was incubated with CLCP fractions and *H. pylori* suspension at the same time for 2 h. After incubation, the medium was replaced with a freshly prepared serum-free RPMI 1640 medium containing no antibiotics; (c) in the post-incubation experiment, a monolayer was infected with *H. pylori* suspension for 2 h, then washed three times with PBS. Next, the CLCP fractions were added and cultured. The multiplicity of infection (M.O.I) for all the experiments was 100:1. At the end of incubation, the cell supernatants were collected, while cells were washed five times with PBS, and then digested with 500 μL trypsin. The digested solution was mixed with two volumes of urease test solution for 2 h and then the absorbance at 550 nm was measured to evaluate anti-adhesive activity.

### 2.8. Evaluation of Inflammatory Responses in AGS Cells

AGS cells were plated at 4 × 10^5^ cells/well in 24-well plates and then a HP 700392 *H. pylori* suspension was added for 2 h before co-incubation with CLCP fractions for 24 h. AGS cells cultured in the absence of HP 700392 *H. pylori* suspension and CLCP fractions served as a control. Then, the supernatant medium of 100 μL in each group was collected to measure human IL-8 levels using a sandwich ELISA kit (ab214030, Abcam, Cambridge, MA, USA), according to the manufacturer’s instructions. AGS-treated cells were washed three times with PBS, then extracted using a ReadyPrep™ protein extraction kit (163-2086, Bio-Rad, Hercules, CA, USA) to collect total proteins. The amounts of secreted nuclear factor kappa B (NF-κB) p65 and beta-actin (β-actin) were quantified by using commercially available ELISA kits, Abcam’s NFκB p65 in vitro SimpleStep ELISA™ (ab176648, Abcam, Cambridge, MA, USA) and total β-Actin Sandwich PathScan^®^ ELISA (#7881, Cell Signaling, Danvers, MA, USA).

### 2.9. Analysis of Gene Expression by Quantitative Reverse Transcription Polymerase Chain Reaction

AGS cells were seeded at 4 × 10^5^ cells/well in 24-well plates and then a HP 700392 *H. pylori* suspension was added for 2 h before co-incubation with CLCP fractions for 24 h. The supernatant was discarded, and total RNA was extracted from the treated AGS cells with TRIzol reagent (Invitrogen, Carlsbad, CA, USA) according to the manufacturer’s protocol. RNA was further reverse transcribed using the iScript™ cDNA synthesis kit (Bio-Rad Laboratories, Hercules, CA, USA), and qRT-PCR was performed in a SYBR Premix Ex Taq II (Takara-Bio, Tokyo, Japan) under the following conditions: denaturation for 1 min at 95 °C, annealing for 20 s at 60 °C, and extension for 30 s at 72 °C for 30 cycles. For amplification of human IL-8, NF-κB p65, glyceraldehyde-3-phosphate dehydrogenase (GAPDH), the specific primers of protein-coding genes are *IL-8* forward primer: ACA CTG CGC CAA CAC AGA AAT TA, reverse primer: TTT GCT TGA AGT TTC ACT GGC ATC; *NF-κB p65* forward primer: ACG ATC TGT TTC CCC TCA TCT, reverse primer: TGG GTG CGT CTT AGT GGT ATC; *GAPDH* forward primer: TGT GTC CGT CGT GGA TCT GA, reverse primer: TTG CTG TTG AAG TCG CAG GAG. *IL-8* and *NF-κB p65* gene expressions, relative to GAPDH, were calculated according to the 2^−ΔΔCp^ method [29].

### 2.10. Statistical Analysis

All experiments were performed in triplicate at least. The data were expressed as the mean ± SD. Statistical significance was evaluated using one-way ANOVA, followed by Tukey’s test with GraphPad Prism (GraphPad Prism for Windows 9.0; GraphPad, San Diego, CA, USA) with *p* values lower than 0.05 being considered statistically significant. Principal components analysis was performed by using the R-based software Chemometric Agile Tool (CAT) developed by the Italian group of Chemometrics [30].

## 3. Results

### 3.1. Yields and Compositions of the Sulfate Polysaccharide from Caulerpa lentillifera

The crude polysaccharides were obtained through hot-water extraction, followed by ethanol precipitation, deproteinization, and lyophilization, by which the yield of CLCP was estimated at about 8.93% ± 0.56%. CLCP was purified by using DEAE-Sephadex A25 to obtain CLCP-1 and CLCP-2 with a yield of 4.19% and 6.27%, respectively (Figure 1).

The total sugar, protein, uronic acid, and sulfate content of CLCP and its fractions were determined and are shown in Table 1. The three CLCP contained comparable contents of total sugar (65.51–87.51%) and protein (0.24–2.94%), and varied contents of uronic acid (2.81–12.48%). The polysaccharide from *C. lentillifera* is a sulfate polysaccharide, in which the sulfate content of CLCP, CLCP-1, and CLCP-2 was 8.11, 16.21%, and 18.15%, respectively, indicating that they were sulfated polysaccharides. The Mw of CLCP-1 and CLCP-2 was estimated as 963.15 kDa, and 648.42 kDa, respectively. The monosaccharide analysis of the fractions obtained after CLCP–hydrolysis indicated that CLCP –1 contains a neutral monosaccharide composition of mannose (46.85%), galactan (22.78%), xylose (20.81%), and glucose (2.18%) with a molar ratio of 21.5:8.4:3.2:1.0; and CLCP-2 was comprised mannose (50.21%), galactan (24.86%), xylose (23.21%), and glucose (1.85%) with a molar ratio of 27.1:13.4:12.5:1.0.

Figure 2A shows the infrared spectra of the three CLCPs’ polysaccharide fractions. All three CLCPs displayed quite similar and typical characteristics of sulfated polysaccharide profiles. Principal component analysis (PCA) was used to visualize the differences among the CLCPs’ polysaccharide spectra. Figure 2B shows the scores obtained by the CLCPs for the first two principal components (PC1 and PC2). As can be seen in Figure 2B, PC1 explains 81.6% of the variability of the data, while PC2 explains 18.4% of the variance.

### 3.2. In Vitro Antimicrobial Potential of CLCP against H. pylori

Figure 3 shows the antimicrobial effects induced by CLCPs against four tested *H. pylori* strains in Brucella broth supplemented with sheep blood with various concentrations of CLCP-1 or CLCP-2 up to 1000 µg/mL in microaerophilic condition with moderate agitation at 37 °C. A dose- and strain-dependent reduction of *H. pylori* growth induced by CLCPs were observed. As seen in Figure 3, the highest antimicrobial effects are related to the concentrations of CLCP-1 and CLCP-2 tested. Moreover, the different structures of CLCP significantly affected (*p* ≤ 0.05) the antimicrobial effect exerted against *H. pylori* under the same culture condition and the same concentrations of CLCPs applied. Among four tested *H. pylori* strains, HP 700392 showed susceptibility to CLCPs, so it was chosen to further investigate the effects.

### 3.3. Effect of CLCP on Cell Viability

The different concentrations (31.25–1000 µg/mL) of CLCP-1 or CLCP-2 were treated with AGS cells to assess the toxicity (Figure 4A). In addition, RAW 264.7 cell were treated with different concentrations (312.5–1000 µg/mL) of CLCP-1 or CLCP-2 in the absence or presence of LPS (1 µg/mL). The results indicate that CLCP-1 and CLCP-2 do not show any cytotoxic effect on the viability of AGS and RAW 264.7 cells. As shown in Figure 4B, the viability of LPS-stimulated RAW 264.7 cells decreased to 87.3% compared to the cells in the non-treated group and LPS (control group, 100%). In contrast, the cell proliferation was not significantly affected by CLCP-1 and CLCP-2 in the presence of LPS up to 1000 µg/mL for 24 h. Based on these data, the concentrations of CLCP-1 and CLCP-2 (62.5–1000 µg/mL) were used for the subsequent experiments.

### 3.4. The Effect of CLCP on the Nitric Oxide Production in RAW 264.7 Cells

To probe the effect of CLCP on nitric oxide (NO) inhibition, the RAW 264.7 cells were treated with LPS to induce the level of NO production, while the different concentrations of CLCP-1 or CLCP-2 were added to the induced-RAW 264.7 cells. The results showed that the production of NO achieved a significant difference for CLCP-1 or CLCP-2 at 62.5 μg/mL in the cells compared to that of the LPS group. (Figure 5A). As shown in Figure 5B, LPS significantly stimulated the expression of iNOS protein in RAW 264.7 cells. However, the expression of iNOS in RAW 264.7 cells was effectively suppressed by CLCP-1 or CLCP-2 treatment in a concentration-dependent manner (Figure 5B). These results indicate that CLCP-1 and CLCP-2 protected RAW 264.7 cells against LPS-stimulated cell death by inhibiting iNOS protein production.

### 3.5. Inhibition of H. pylori Infection on AGS Cells by CLCPs

We further assessed the inhibitory activity of the CLCPs against *H. pylori* infection in AGS cells. The CLCPs were tested under three conditions: post-treatment, pre-treatment, and co-treatment at a multiplicity of infection (MOI) of 100 (Figure 6). A significant reduction in *H. pylori* adhesion was observed in cells post-treatment with CLCP-1 or CLCP-2 at 500 μg/mL in comparison to that of the untreated group. When the cells were treated with 1000 μg/mL of CLCP-1 or CLCP–2, about a 35% reduction in the infection rate was observed in the post-treatment of cells while it had no inhibitory activity in the other two groups, suggesting that the CLCPs directly interact with *H. pylori*.

### 3.6. The Activity and Mechanism of CLCP in Reducing H. pylori Infection

*H. pylori* significantly increased the levels of IL-8 in AGS cells, whereas CLCP-1 and CLCP-2 significantly (*p* < 0.001) suppressed IL-8 production (up to 38.1% and 38.3%, respectively) in *H. pylori*-stimulated AGS cells as expected (Figure 7). qRT-PCR was used to determine the effects of CLCP-1 and CLCP-2 on IL-8 gene transcription. Similarly, *H. pylori* treatment significantly increases the expression levels of IL-8, compared to the *H. pylori*-untreated group (Figure 7C). In the presence of CLCP-1 and CLCP–2, IL-8 mRNA levels were downregulated by 57.7% and 59.0%, compared with the *H. pylori* treatment. The activity of NF-κB, a trigger for the production of IL-8, was 3.1-fold higher in the *H. pylori* treatment compared to *H. pylori*-untreated group. Additionally, the mRNA expression level of NF-κB in CLCP-1 and CLCP-2 (500 and 1000 μg/mL) was significantly inhibited in *H. pylori*-stimulated AGS cells in a dose-dependent manner.

## 4. Discussion

The extraction yield of polysaccharide is comparable to that of *C. lentillifera* polysaccharides extracted in previous studies (Long, 2020; Zhang, 2020) and is higher than red seaweeds (Tan, 2020). Recently, the sulfate levels of *C. lentillifera* were reported to range between 1.9 and 22.7% (Sun, 2018; Long, 2020). Moreover, polysaccharides rich in sulfate groups have been reported to have better bioactivities (Long, 2020). The Mw of these two polysaccharides is smaller than those obtained by Sun et al. with an average molecular weight of 2589.9–4068.5 kDa [22]. Since sulfated mannose are major polysaccharide components of *C. lentillifera* [14], mannose was found to be one of the dominant sugar units for all of the polysaccharide samples.

Interaction of the sulfate polysaccharide CLCPs with the bacterial surface is expected to depend mainly on the charge of the polymer in many cases. The overall net surface charge of the CLCPs is related to their surface potential. The ζ-potential of three CLCPs are negatively charged owing to the high presence of uronic acid and sulfated groups in its structure, which is similar to what is expected from polysaccharide fractions from *Sargassum fusiforme* [31]. The higher zeta potential indicated better stability.

The broad absorbance peaks at around 3400 cm^−1^ and 2930 cm^−1^ indicate the O–H and C–H stretching vibration of the polysaccharide and the region of 1030–1044 cm^−1^ was probably associated with C–O–H moieties of glucosides or C–O–C stretching vibrations in the rings [32,33]. The bands near 1250 cm^−1^ and 1440 cm^−1^ were due to the S–O stretching vibrations, which positively correlated with the sulfate content [33,34]. The relative strong absorption band near 1655 cm^−1^ may be from asymmetrical stretching vibrations of the C=O bond in the uronic acid or the protein amide I [35]. The protein contents in CLCP-1 and CLCP-2 are much lower than their uronic acid contents, so the carbonyl groups in uronic acid are the main contributors to this peak. Strong distinguishable bands at the 1540 cm^−1^ range that correspond to the protein [36] observed in CLCP but not to the protein in CLCP-1 and CLCP-2 might explain the probable loss of protein during extraction. Principal component analysis (PCA) is a mathematical technique for reducing the dimensionality of large datasets into a few orthogonal PCs from hundreds of spectral data points [37]. Thus, it is especially useful in the interpretation of the FT-IR spectra of polysaccharides, which are diverse and complicated depending on their origin or production methods [38]. As shown in the PCA, PC1 allows for distinction between CLCP-2 and other CLCPs. Moreover, CP2 allows for distinction between CLCP-1 and CLCP. This result relays some significant differences between the structures of CLCPs.

Sulfate polysaccharides were reported to be effective in bactericidal activity against a wide variety gram-positive and gram-negative bacterium, especially multi-resistant strains [39]. Furthermore, numerous sulfate polysaccharides derived from seaweed suppressed the process of *H. pylori* adhesion to the gastric mucosa through in vitro and in vivo tests [40]. Previous studies have found that the antiadhesive activity against *H. pylori* of fucoidan (2000 μg/mL) was 40% [41]. Another study of experimental fluorescent-labeled *H. pylori* J99 to human AGS cells treated with arabinogalactan protein (BA1) from *Basella alba* stem showed that the high dose of AB1 (2 mg/mL) could markedly block the adhesion of ~67% of the tested *H. pylori* isolates [42]. In case of multiple paraffin-embedded tissue sections from human gastric mucosa, a 2 h pre-treatment of the bacteria with raw polysaccharides from licorice roots (*Glycyrrhiza glabra* L.) (1 mg/mL) reduced the bacterial binding by 60% [43]. Our present study confirmed the results of previous studies and proved that sulfate polysaccharides from *C. lentillifera* have appropriate anti-*H. pylori* effects.

The increased NO levels in macrophages triggered deleterious consequences such as chronic inflammation, carcinogenesis, and sepsis [44]. Notably, macrophages upregulated the inducible nitric oxide synthase (iNOS) transcription to generate NO from L-arginine during the pathogenesis of *Helicobacter pylori* [45], as the intracellular presence of excessive NO production led to constant gastric mucosal disturbance, chronic gastritis, and a multi-step complex pathway and process that initiates gastrointestinal tumor formation. In this study, we verified that the level of CLCP-1 and CLCP-2 significantly decreased NO production and downregulated iNOS mRNA expression in LPS-activated RAW 264.7 cells after CLCP-1 and CLCP-2 treatment. It is also acknowledged that increased NO levels were related to sulfated content and the proportion of monosaccharides in polysaccharides. Sulfated polysaccharides isolated from Codium fragile contain 21.06% sulfate with high inhibitory inflammation activity [46]. Similarly, we observed that *Saccharina japonica* polysaccharides rich in sulfate groups (30.72%) also showed strong anti-inflammatory effects on LPS-induced RAW 264.7 cells via suppressing the phosphorylation of MAPK and NF-κB [47]. These results suggest the anti-inflammatory effect of *C. lentillifera* due to its high sulfate and mannose.

*H. pylori* induces various signaling pathways resulting in the phosphorylation of transcription factors NF-κB and activator protein-1 (AP-1) and the downstream transcription of IL-8 in gastric epithelial cells [48,49]. Upon post-treatment with the CLCP-1 and CLCP-2 inhibitors, the capability of *H. pylori* to release pro-inflammatory cytokine IL-8 and NF-κB decreased in the bacteria-infected AGS cells. Together with previous work, the findings presented here indicate that CLCP should be considered in efforts to develop a promising agent to reduce the adhesion capability of *H. pylori* to host cells as well as to inhibit the induction of inflammation responses in host cells.

The most prominent feature of sulfate polysaccharides from *Caulerpa lentillifera* is their rich and diverse structure and bioavailability. Two polysaccharide fractions (CLCP-1 and CLCP-2) were extracted from *C. lentillifera* with a molecular weight < 100 kDa for the first time. Structural data indicated that CLCP-1 and CLCP-2 have many similar structural features while differing in their molecular weights and mono-sugar ratios. In particular, CLCP-1 and CLCP-2 have displayed superior antimicrobial activity against *H. pylori* and low toxicity. The antibacterial effects of CLCPs against H. pylori are shown through their prohibition of *H. pylori* adhesion to host cells by promoting the secretion of IL-8 and NF-κB. Moreover, CLCP-1 and CLCP-2 significantly inhibited NO production by affecting the expression of iNOS mRNA in vitro. To the best of the authors’ knowledge, this is the first comprehensive study to have shown that CLCP-1 and CLCP-2 exhibit a potent protective effect against *H. pylori*-infected gastric adenocarcinoma cells.

## Figures and Tables

**Figure 1 polymers-14-04993-f001:**
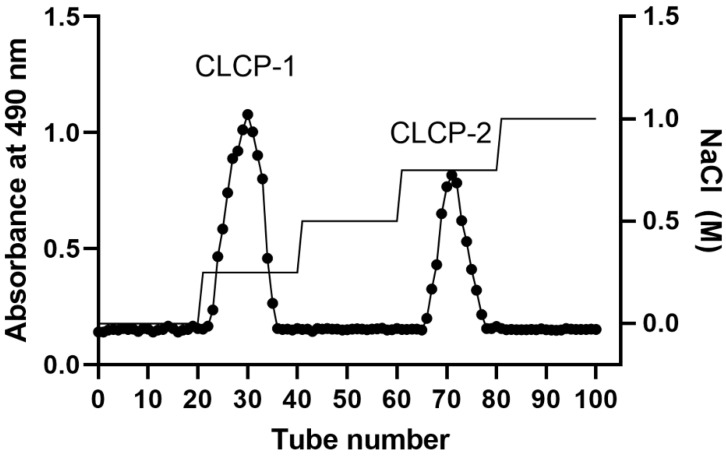
Purification of *Caulerpa lentillifera* polysaccharides fractions using DEAE-Sephadex A-25.

**Figure 2 polymers-14-04993-f002:**
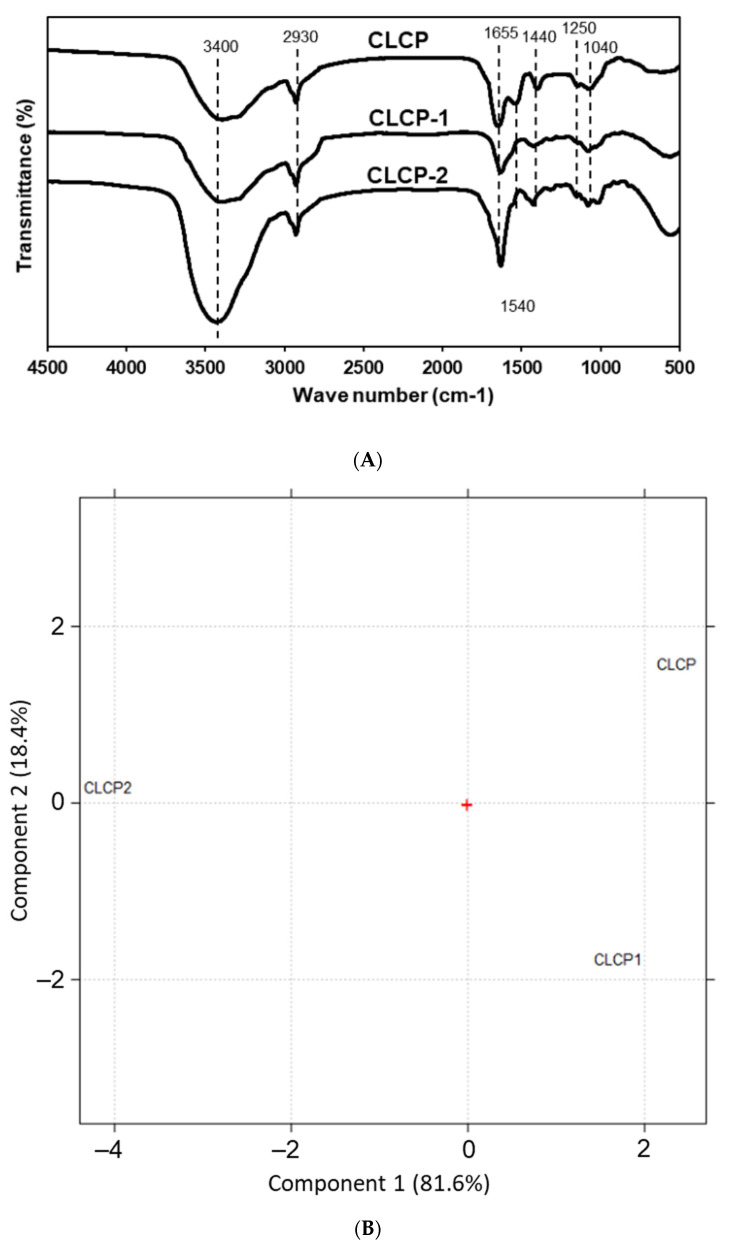
FT-IR spectra (**A**) and PCA analysis (**B**) of CLCP, CLCP-1, and CLCP-2.

**Figure 3 polymers-14-04993-f003:**
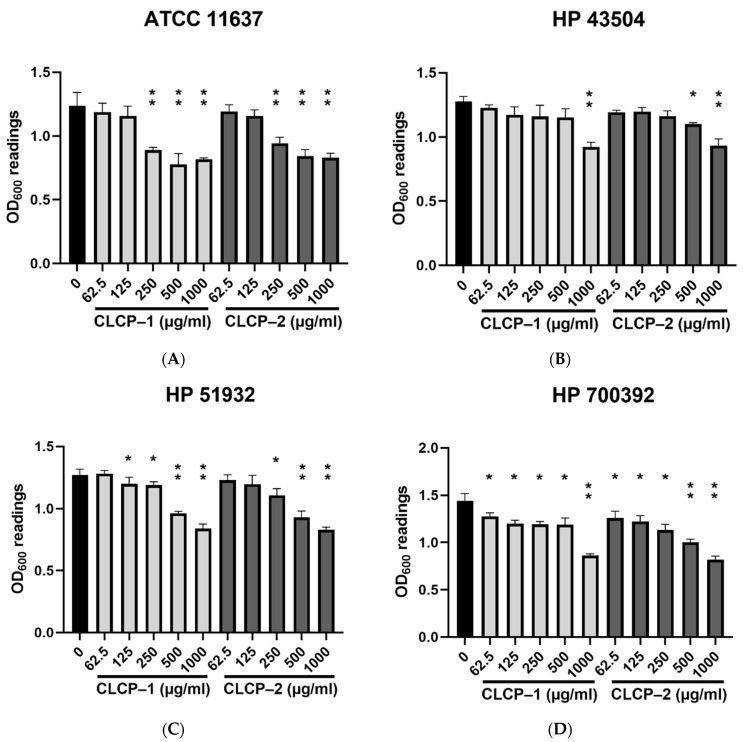
Antimicrobial activity of CLCP-1 and CLCP-2 against *H. pylori* ATCC11637 (**A**), HP43504 (**B**), HP 51932 (**C**), and HP 700392 (**D**). The values are expressed as the means ± SD with * *p* < 0.05, ** *p* < 0.01 compared with the control group, *n* = 3.

**Figure 4 polymers-14-04993-f004:**
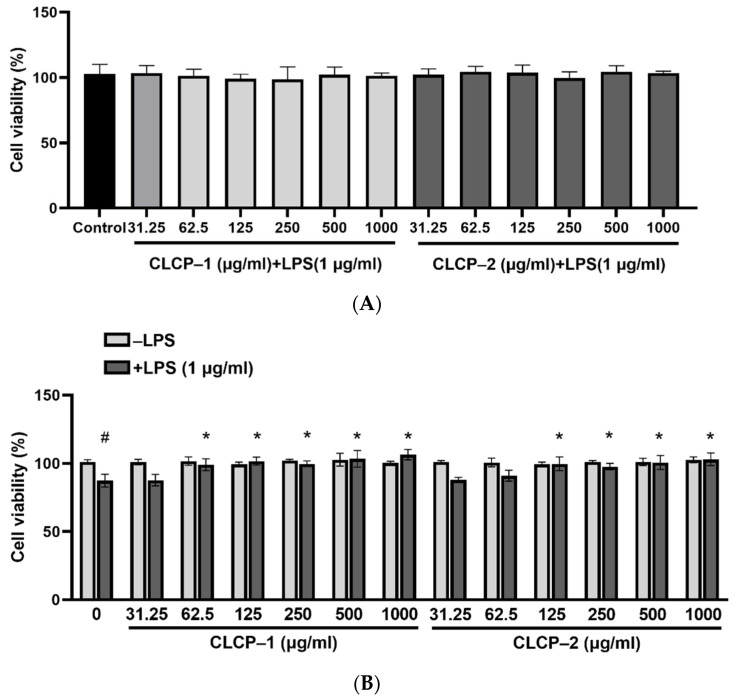
Measurement of the potential cytotoxicity of CLCP-1 and CLCP-2 in RAW 264.7 cells (**A**) an AGS cells (**B**) by MTT assay. The values are expressed as the means ± SD with ^#^
*p* < 0.05 compared with the group without LPS; * *p* < 0.05 compared with the presence of LPS group, *n* = 3.

**Figure 5 polymers-14-04993-f005:**
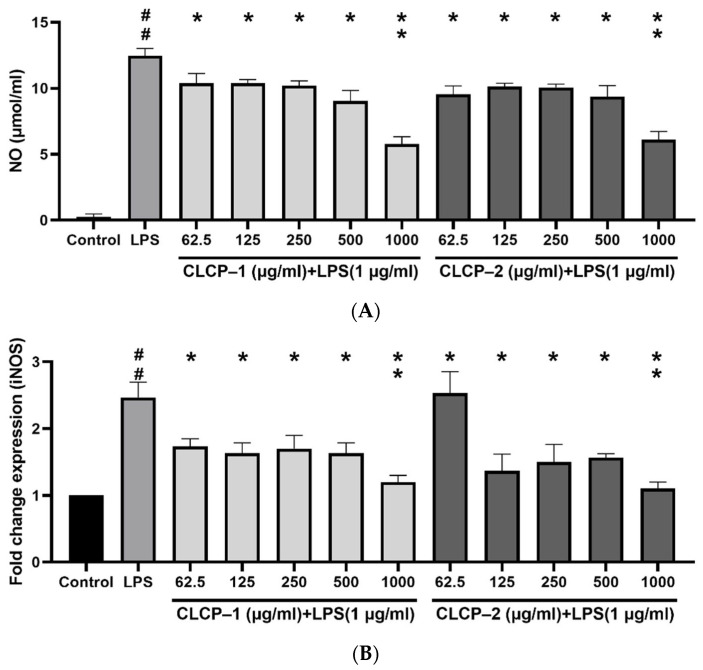
Effect of CLCP-1 and CLCP-2 on NO production (**A**) and iNOS mRNA expression (**B**) in LPS-activated RAW 264.7 cells. The values are expressed as the means ± SD with ^##^
*p* < 0.05 compared with the control group; * *p* < 0.05, ** *p* < 0.01 compared with the LPS group, *n* = 3.

**Figure 6 polymers-14-04993-f006:**
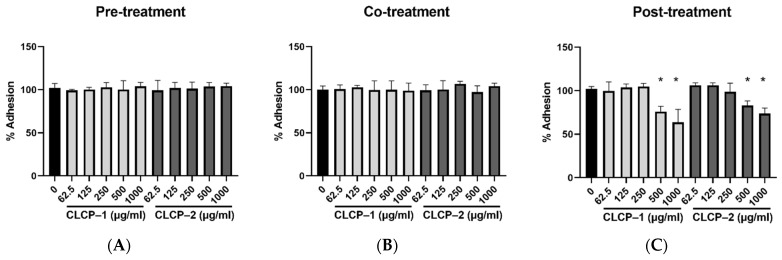
Inhibitory activity of CLCP-1 and CLCP-2 in *H. pylori*-infected AGS cells under pre-treatment (**A**), co-treatment (**B**), and post-treatment (**C**). The values are expressed as the means ± SD with * *p* < 0.05 compared with the control group, *n* = 3.

**Figure 7 polymers-14-04993-f007:**
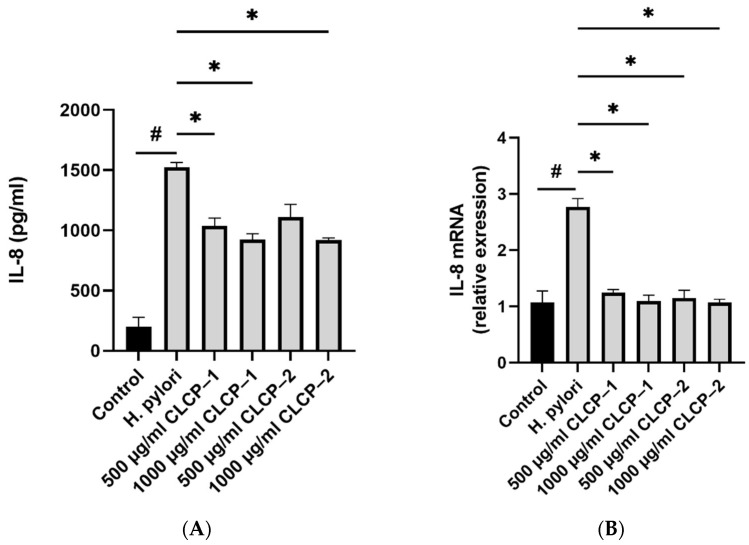
CLCP-1 and CLCP-2 inhibit IL-8 and NF-κB mRNA and protein levels in *H. pylori*-infected AGS cells. (**A**) IL-8 levels measured by ELISA detection. (**B**) IL-8 transcription measured by qRT-PCR. (**C**) NF-κB levels measured by ELISA detection. (**D**) NF-κB transcription measured by qRT-PCR. ^#^
*p* < 0.05 compared with the control group; * *p* < 0.05, ** *p* < 0.01, *** *p* < 0.001, **** *p* < 0.0001 compared with the *H. pylori* group, *n* = 3.

**Table 1 polymers-14-04993-t001:** Chemical properties of the *C. lentillifera* polysaccharide.

	CLCP	CLCP-1	CLCP-2
Yield (%)	8.93 ± 0.56	1.38 ± 0.17	1.03 ± 0.08
Chemical characteristics (%)			
Total sugar	87.51 ± 2.13	72.19 ± 1.51	65.51 ± 3.04
Protein	2.94 ± 0.02	0.24 ± 0.02	0.35 ± 0.07
Uronic acid	12.48 ± 1.08	5.36 ± 0.48	2.81 ± 0.64
Sulfate	8.11 ± 0.24	16.21 ± 0.29	18.15 ± 1.95
Mw (kDa)	NA ^1^	963.15	648.42
Monosaccharide composition (%)			
Man	38.18	46.85	50.21
Xyl	11.23	20.81	23.21
Gal	29.31	22.78	24.86
Glc	3.51	2.18	1.85
z-potential (mV)	−15.6 ± 0.2	−22.61 ± 0.5	−26.1 ± 0.9

^1^ NA, not applicable.

## Data Availability

Not applicable.

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
