# Peer review of "Preparation, Characterization, and Anti-Adhesive Activity of Sulfate Polysaccharide from Caulerpa lentillifera against Helicobacter pylori"

_polymers, 2022, doi:10.3390/polym14224993_

Round 1

Reviewer 1 Report

On request of Polymers, I have revised the manuscript titled “Preparation, characterization, and anti-adhesive activity of sulfate polysaccharide from Caulerpa lentillifera against Helicobacter pylori”, by Bao and colleagues.

The main scope of this study was to investigate the possible anti-adhesive and consequently antibacterial activity of two low molecular sulfate polysaccharides present in the aqueous extracts of C. lentillifera (CLCP), since its polysaccharides have been already reported to possess broad antimicrobial activity and anti-inflammatory potentials. To this end, after hot water extraction and purification process, two purified poly-saccharide fractions (CLCP-1 and CLCP2) were structurally characterized and biologically evaluated against H. pylori. Experiments to investigate the possible mechanism of action of the antibacterial and anti-adhesive effects of CLCP-1 and CLCP-2 have been also included in this study.

Considering, the high incidence of gastric tumor, associated to the gastric mucosa, chronic inflammation due to Helicobacter pylori infection, the present study concerning CLCP1 and CLCP-2 could be very interesting. Anyway, there are some major and minor issues which must be addressed by authors, before further consideration of their manuscript.

1)      Introduction is rather poor. The authors should provide more detail concerning H. pylori, the H. pylori gastric infections, the clinically approved available antibiotics, their chemical structures, the molecules already reported in literature with potentialities as new antibacterial agents against H. pylori, their advantages, and disadvantages, such as reported cytotoxicity, etc.

2)      The discussion of the FTIR spectra is very poor, and should be improved. I suggest authors to include the spectral data obtained for the three samples of CLCP in a data-set matrix, and to process it using the principal components analysis (PCA). To process spectral data by PCA, the authors could use CAT statistical software, (Chemometric Agile Tool, free down-loadable online, at: http://www.gruppochemiometria.it/index.php/software/19-download-the-r-based-chemometric-software; accessed on October, 23, 2022). An extensive discussion on PCA results sould be included in this study.

3)      My most concern is about the numbers of the active concentrations of both CLCP-1 and CLCP-2. In the most favorable case of HP 700392, CLCP-1 and -2 showed significant attivity at concentrations > 1000 µM. Authors should discuss better and in details this findings highlighting while, although such high active concentrations, CLCP1 and 2  could anyway represent accellent candidates to develop new antibacterial agents to treat infections sustained by H. pylori

As minor issues, I suggest authors checking well all manuscript to detect typos or grammatical errors (in the keywords use the italics when necessary, line 55 remove “and”, line 57 correct CLCS in CLCP, etc.) and to adapt better the manuscript to Polymers template (check the format of subheadings, use Figure and not FIG.).

I will reconsider this manuscript after major revisions.

Author Response

On request of Polymers, I have revised the manuscript titled “Preparation, characterization, and anti-adhesive activity of sulfate polysaccharide from Caulerpa lentillifera against Helicobacter pylori”, by Bao and colleagues.

The main scope of this study was to investigate the possible anti-adhesive and consequently antibacterial activity of two low molecular sulfate polysaccharides present in the aqueous extracts of C. lentillifera (CLCP), since its polysaccharides have been already reported to possess broad antimicrobial activity and anti-inflammatory potentials. To this end, after hot water extraction and purification process, two purified poly-saccharide fractions (CLCP-1 and CLCP2) were structurally characterized and biologically evaluated against H. pylori. Experiments to investigate the possible mechanism of action of the antibacterial and anti-adhesive effects of CLCP-1 and CLCP-2 have been also included in this study.

Considering, the high incidence of gastric tumor, associated to the gastric mucosa, chronic inflammation due to Helicobacter pylori infection, the present study concerning CLCP1 and CLCP-2 could be very interesting. Anyway, there are some major and minor issues which must be addressed by authors, before further consideration of their manuscript.

Response: We are very grateful for providing comments for our manuscript. The parts pointed out by the reviewer were carefully corrected and add the explanation to support the revision.

1)      Introduction is rather poor. The authors should provide more detail concerning H. pylori, the H. pylori gastric infections, the clinically approved available antibiotics, their chemical structures, the molecules already reported in literature with potentialities as new antibacterial agents against H. pylori, their advantages, and disadvantages, such as reported cytotoxicity, etc.

Response: Thank you for your comment. According to your suggestions, we revised the introduction to improve the knowledge about Helicobacter pylori eradication regimen as below.

Line 35-47:

Despite elimination rates of 60% to 90%, some concerns persist as the rise of antibiotic re-sistance (with high divergence) and the effectiveness of current regimens have diminished over the years. Since 2017, the World Health Organization (WHO) announces that the prevalence of resistance of H. pylori to clarithromycin and metronidazole is over 15% [4]. Clarithromycin is the second-generation macrolide and the most potent antibiotic in H. pylori eradication treatment regimens [5]. Clarithromycin inhibits the protein synthesis of H. pylori by targeting the 50S ribosomal subunit [6]. Metronidazole is a synthetic nitroim-idazole derivative activated by nitro-reductase to produce oxygen radicals toxic to bacteria through DNA damage [7]. Metronidazole-resistant H. pylori may be caused by mutations in rdxA, which encodes oxygen-insensitive NADPH nitroreductase [8]. Following the rapid development of antibiotic resistance, the other drawbacks of these treatment failures are antibiotic degradation by the acidic stomach environment, use of non-FDA-approved agents (e.g. tetracyclin and nitazoxanide), severe adverse effects, and poor patient com-pliance [9,10]

2)      The discussion of the FTIR spectra is very poor, and should be improved. I suggest authors to include the spectral data obtained for the three samples of CLCP in a data-set matrix, and to process it using the principal components analysis (PCA). To process spectral data by PCA, the authors could use CAT statistical software, (Chemometric Agile Tool, free down-loadable online, at: http://www.gruppochemiometria.it/index.php/software/19-download-the-r-based-chemometric-software; accessed on October, 23, 2022). An extensive discussion on PCA results sould be included in this study.

Response: We appreciate your comment and have agreed your opinion. We downloaded and used CAT to analyse FTIR spectra. We have revised in detail as below.

Line 214-215 – Materials and Methods:

Principal components analysis was performed by using the R-based software Chemometric Agile Tool (CAT) developed by the Italian group of Chemometrics [30].

Line 241-243 – Results:

Principal component analysis (PCA) was used to visualize the differences among the CLCPs polysaccharide spectra. As shown in Figure 2B, the total change of PC1 and PC2 was 81.6% and 18.4% of the variation, respectively.

Line 338-346 – Discussion:

Principal component analysis (PCA) is a mathematical technique for reducing the dimensionality of large datasets into a few orthogonal PCs from hundreds of spectral data points into a few orthogonal PCs [37]. Thus, it is especially useful in the interpretation of FT-IR spectra of polysaccharides, which are diverse and complicated depending on their origin or production methods [38]. As shown on the PCA, the CLCP and CLCP-1 were positioned to the right along the PC1 axis with positive loading, not for CLCP-2. However, CLCP lay on the positive side of PC2, while CLCP-1 was on the negative side of PC2. Therefore, the different CLCPs were well separated.

3)      My most concern is about the numbers of the active concentrations of both CLCP-1 and CLCP-2. In the most favorable case of HP 700392, CLCP-1 and -2 showed significant attivity at concentrations > 1000 µM. Authors should discuss better and in details this findings highlighting while, although such high active concentrations, CLCP1 and 2 could anyway represent accellent candidates to develop new antibacterial agents to treat infections sustained by H. pylori

Response: Thank you for your useful comment for our study. We added some previous studies to support our hypothesis. The text has been revised to be more detailed as below.

Line 351-358:

Previous studies have found that the antiadhesive activity against H. pylori of fucoidan (2000 μg/mL) was 40% [41]. Another study of experimental fluorescent-labeled H. pylori J99 to human AGS cells treated with arabinogalactan protein (BA1) from Basella alba stem that the high dose of AB1 (2 mg/mL) could markedly block the adhesion of ~67% of the tested H. pylori isolates [42]. In case of multiple paraffin-embedded tissue sections from human gastric mucosa, a 2 h pre-treatment of the bacteria with raw polysaccharide from Liquorice roots (Glycyrrhiza glabra L.) (1 mg/mL) reduced the bacterial binding by 60% [43]. Our present study confirmed previous studies and proved that sulfate polysaccharide from C. lentillifera has appropriate anti-H. pylori effect.  

4) As minor issues, I suggest authors checking well all manuscript to detect typos or grammatical errors (in the keywords use the italics when necessary, line 55 remove “and”, line 57 correct CLCS in CLCP, etc.) and to adapt better the manuscript to Polymers template (check the format of subheadings, use Figure and not FIG.).

I will reconsider this manuscript after major revisions.

Response: We appreciate your comments and apologize again for the mistakes. We revised these typos carefully.

Reviewer 2 Report

It was a great pleasure to read this manuscript. I deem that the manuscript can be published in Polymers in its current form.

Author Response

We would like to thank the reviewers for their thoughtful comments and efforts towards improving our manuscript.

Round 2

Reviewer 1 Report

Dear Authors,

I appreciate your work, expecially for what concerns the discussion of the FTIR results using the chemometric tool I suggested you. Unfortunatelly, although CAT has been used correctly and the reported plot is OK, many expressions you used in the description of the results and in the discussion are not correct. As exmples, it is not correct saying that the percentages associated to PCs are "total chenges". The numbers 82% and 18% represent the percentage of variance explained by PC1 and PC2 respectively as shown in the scree plot provided by CAT. Additionally, the coordinates of samples in the space of the new variables, i.e. the PCs, are the scores not the loadings. The loading plot is another type of plot. The author must correct both te results section and the discussion one. Ref. "Biodegradable and Compostable Shopping Bags under Investigation by FTIR Spectroscopy. Appl. Sci. 202111, 621. https://doi.org/10.3390/app11020621, could be of help for you.

Best

Author Response

Thank you for pointing out this issue. We revised the PCA analysis below:
Line 245-248, Results
Principal component analysis (PCA) was used to visualize the differences among the CLCPs polysaccharide spectra. Figure 2B shows the scores obtained by the CLCPs for the first two principal components (PC1 & PC2). As can be seen in Figure 2B, PC1 explains 81.6% of the variability of the data, while PC2 explains 18.4% of the variance.
Line 347-350, Discussion
As shown on the PCA, PC1 allows to distinguish between CLCP-2 and other CLCPs. Besides, CP2 allows distinguishing between CLCP-1 and CLCP. This result announces some significant differences between the structures of CLCPs.